# Optimizing marine macrophyte capacity to locally ameliorate ocean acidification under variable light and flow regimes: Insights from an experimental approach

**Aurora M. Ricart**[1]*, **Brittney Honisch**[1], **Evangeline Fachon**[2], **Christopher W. Hunt**[3], **Joseph Salisbury**[3], **Suzanne N. Arnold**[4], **Nichole N. Price**[1]

**1** Bigelow Laboratory for Ocean Sciences, East Boothbay, Maine, United States America, **2** Massachusetts Institute of Technology, Woods Hole Oceanographic Institution, Woods Hole, Massachusetts, United States of America, **3** Ocean Process Analysis Laboratory, University of New Hampshire, Durham, NH,, United States of America, **4** Island Institute, Rockland, Maine, United States of America

* aricart@bigelow.org

**Data Availability Statement:** All relevant data are within the paper and its Supporting Information files.

## Abstract

The urgent need to remediate ocean acidification has brought attention to the ability of marine macrophytes (seagrasses and seaweeds) to take up carbon dioxide ($CO_2$) and locally raise seawater pH via primary production. This physiological process may represent a powerful ocean acidification mitigation tool in coastal areas. However, highly variable nearshore environmental conditions pose uncertainty in the extent of the amelioration effect. We developed experiments in aquaria to address two interconnected goals. First, we explored the individual capacities of four species of marine macrophytes (*Ulva lactuca*, *Zostera marina*, *Fucus vesiculosus* and *Saccharina latissima*) to ameliorate seawater acidity in experimentally elevated pCO2. Second, we used the most responsive species (i.e., *S. latissima*) to assess the effects of high and low water residence time on the amelioration of seawater acidity in ambient and simulated future scenarios of climate change across a gradient of irradiance. We measured changes in dissolved oxygen, pH, and total alkalinity, and derived resultant changes to dissolved inorganic carbon (DIC) and calcium carbonate saturation state ($\Omega$). While all species increased productivity under elevated $CO_2$, *S. latissima* was able to remove DIC and alter pH and $\Omega$ more substantially as $CO_2$ increased. Additionally, the amelioration of seawater acidity by *S. latissima* was optimized under high irradiance and high residence time. However, the influence of water residence time was insignificant under future scenarios. Finally, we applied predictive models as a function of macrophyte biomass, irradiance, and residence time conditions in ambient and future climatic scenarios to allow projections at the ecosystem level. This research contributes to understanding the biological and physical drivers of the coastal $CO_2$ system.

**Funding:** This study was supported by NASA (https://www.nasa.gov/) grant NX14AL84G to JS, NOAA (https://www.noaa.gov/) grants N17OAR0170164 to JS & NA17NMF4270202 to NP, the Broad Reach Foundation to NP (https://www.broadreachfoundation.org/), the Nature Conservancy to NP (https://www.nature.org/), and the NSF REU Program to NP (https://www.nsf.gov/crssprgm/reu/) (grants 1156740 and 1460861). The funders had no role in study design, data collection and analysis, decision to publish, or preparation of the manuscript.

**Competing interests:** The authors have declared that no competing interests exist.

## Introduction

Ocean acidification is among the largest global threats to marine organisms and the ecosystem services they provide [1–3]. The term ocean acidification describes the increase of hydrogen ion ($H^+$) concentration due to rising anthropogenic atmospheric carbon dioxide ($CO_2$) emissions. Absorption of atmospheric $CO_2$ into the oceans' surface induces subsequent changes in the seawater carbonate system, including changes in the concentrations of dissolved inorganic carbon (DIC) forms: increasing dissolved aqueous carbon dioxide ($CO_2$) and bicarbonate ($HCO_3^-$), and decreasing carbonate ions ($CO_3^{-2}$), which result in reduced pH and saturation state of calcium carbonate minerals ($\Omega$) [4, 5]. Additionally, in coastal areas, freshwater riverine inputs and excess nutrient run-off from land can also alter seawater carbonate chemistry and contribute to the increase in $CO_2$ and seawater acidity [6]. These physicochemical changes can differentially and interactively affect marine biota with deleterious consequences for many ecologically and economically important species [7, 8]. The decrease in $CO_2$ concentration and the amelioration of seawater acidity is urgent.

Marine macrophyte species (seagrasses and seaweeds) remove $CO_2$, and some also remove $HCO_3^-$, from seawater through photosynthetic activity while increasing concentration of dissolved oxygen (DO). They remove DIC faster than it can be replaced by diffusion, which alters the speciation of DIC forms and increase pH and $\Omega$ of surrounding waters [9–13]. This ability to modify seawater carbonate chemistry is defined here as "amelioration capacity of seawater acidity". Depending on the extent and strength of this amelioration capacity, marine macrophytes have the potential to create distinct chemical refugia in seawater. Thus, habitat forming species of marine macrophytes could represent a powerful tool for locally mitigating ocean acidification effects, especially in coastal areas [14–16]. However, the multiple biogeochemical and physical processes occurring in coastal zones, coupled with the complexity of carbonate chemistry in marine waters [17], causes high temporal and spatial variability of carbonate parameters [18], and creates uncertainty regarding the generality of potential benefits of the amelioration capacity of seawater acidity of marine macrophytes.

The capacity of marine macrophytes to ameliorate seawater acidity putatively depends on the species biology, the light regime, and hydrodynamic factors [14, 19, 20]. To date, the slow diffusion of atmospheric $CO_2$ in seawater has resulted in an acid-base equilibrium where DIC speciates towards low $CO_2$, low $CO_3^{-2}$, and high $HCO_3^-$ concentrations. Thus, while some macrophytes rely exclusively on the diffusive uptake of $CO_2$ for photosynthesis, many employ carbon concentrating mechanisms to actively increase the concentration of $CO_2$ at the site of photosynthesis [21, 22]. How increasing $CO_2$ will affect the amelioration capacity of macrophyte species might vary depending on whether or not they possess carbon concentrating mechanisms, the affinity of these mechanisms for $HCO_3^-$, and the ability to downregulate them [7, 23, 24]. However, responses to increasing $CO_2$ might also vary with localized environmental conditions affecting species metabolism (e.g. temperature, salinity, nutrients), and, in particular, irradiance and hydrodynamics [25–27]. Light regimes (e.g., daily and seasonal cycles) determine the net carbon uptake of marine macrophytes through photosynthesis and respiration, which in turn can promote fluctuations on the seawater carbonate chemistry parameters [9, 18], while hydrodynamic factors can also influence it by affecting water residence time (i.e., how fast water moves through a system in equilibrium). At high residence time, photosynthetic rates may be relatively low and limited by mass transfer [28, 29], but if submerged vegetation is abundant, the effect of multiple individuals can overlap, resulting in large areas where seawater pH and $\Omega$ are elevated [30].

Overall, the capacity to ameliorate seawater acidity can vary among marine macrophyte species and environmental settings, and the resulting chemical refugia might be limited to

specific locations or periods of time [19, 30, 31]. While photo-physiology of marine macrophytes is generally well understood, how the different drivers of variability on macrophytes amelioration capacity intertwine in the face of increased anthropogenic $CO_2$ emissions is poorly explored. Moreover, most of the available evidence comes from observational correlative and comparative field studies, and there is a lack of evidence from experimental settings in controlled conditions that mimic future projections (but see Anthony *et al*. 2013) [32]. Understanding such relationships will help to elucidate the role of marine macrophytes in a high-$CO_2$ and high-temperature world, and the suitability of marine macrophytes as a mitigation strategy to address the impacts of ocean acidification at local scales.

The aim of this study was to compare among marine macrophyte species to identify which species and under which environmental conditions create chemical refugia to mitigate and adapt to ocean acidification in coastal areas. For that purpose, we developed two sequential experiments in aquaria with high-$CO_2$ derived seawater acidity, and a modeling effort with three main goals: (1) to compare the amelioration capacity of seawater acidity in four ecologically and economically important species of marine macrophytes subjected to increasing levels of $CO_2$ spanning from pre-industrial to predicted future levels in the Representative Concentration Pathways (RCP) 8.5 scenario for the study area [5]. We then used the most responsive species from this comparison (the most promising candidate species for implementation in mitigation strategies) to (2) assess the effects of water residence time on its amelioration capacity in ambient and simulated future scenarios of climate change (increased $CO_2$ and temperature) along a gradient of increasing irradiance. Here we hypothesized that under similar light conditions the amelioration effect will be smaller at low residence time, and that the magnitude of the amelioration effect will be larger in future scenarios of climate change due to more availability of $CO_2$ in seawater. Finally, we used the results from this second experiment to (3) build a model to predict the amelioration effect on seawater acidity as a function of macrophyte biomass, irradiance and water residence time in ambient and future scenarios of climate change to allow projections at the ecosystem level. Throughout the study, we assessed changes in dissolved oxygen (DO), as a proxy of net photosynthetic activity, and dissolved inorganic carbon (DIC), pH and calcium carbonate saturation state ($\Omega$) to assess the amelioration capacity of seawater acidity.

## Materials

### Study species

Four species of marine macrophytes were selected based on their extensive spatial distribution and abundance, the body of literature available on their growth and productivity rates, their important commercial and ecological roles and presence in the area of study, the Gulf of Maine in the east coast of North America. The seagrass *Zostera marina*, commonly known as eelgrass, is a widely distributed temperate species in the northern hemisphere that forms extensive natural monocultures and nursery habitats in the low intertidal and subtidal mudflats [33]. The canopy-forming brown macroalgal species *Saccharina latissima*, commonly known as sugar kelp, is widely distributed in the low intertidal and shallow subtidal cold temperate rocky reefs in the northern hemisphere [34], and represents one of the most farmed seaweed species in the Atlantic ocean [35, 36] with optimum growth of sporophytes reported between 10℃ and 15˚C [37]. *Fucus vesiculosus* is a conspicuous intertidal canopy-forming brown macroalga in the North Atlantic [38], distributed on exposed and protected rocky shores [39], and it is subjected to small-scale wild harvesting in some areas [40]. *Ulva lactuca*, an intertidal green macroalga commonly known as sea lettuce, has a worldwide distribution [41], and is wild harvested or farmed in tank cultures [42, 43]. Each species vary in timing of

peak productivity and relative growth rates. *F. vesiculosus* is a perennial species that overwinters. *Z. marina* can also be perennial but re-establishes vegetative cover in late spring and summer when its productivity is the highest and reproduces in fall. *U. lactuca* and *S. latissima* can reproduce year-round, but also have a seasonal bloom pattern with rapid growth in the late spring and summer. The four species occupy a wide range of habitat conditions from intertidal to shallow subtidal zones, sandy and rocky bottoms, and from sheltered to exposed areas. Collection of marine macrophytes was done under permission from the Department of Marine Resources of the State of Maine (number 2021-53-04).

## Sample collection experimental set-up

**Experiment 1—Comparison among species.** To examine the effects of increased $CO_2$ concentrations upon the capacity of the different species of marine macrophytes to ameliorate seawater acidity, several assays in enclosed well-mixed, airtight chambers with no headspace, were conducted indoors at the Bigelow Laboratory for Ocean Sciences (Maine, USA), in June-July 2015. Thalli of *U. lactuca* and juvenile *S. latissima* were collected as they both began seasonal blooms, together with *F. vesiculosus*. All macroalgal species, free of reproductive structures, were collected from the holdfast at the Bigelow dock in the Damariscotta River Estuary Region (43˚51'38.0"N, 69˚34'41.7"W). Vegetative seagrass shoots of *Z. marina* including 7 cm of rhizome with roots [44] were collected in Broad Cove, Casco Bay (43˚44'56.67"N, 70˚11'37.47"W) also at the peak of productivity. Visually healthy specimens free from visible epiphytes were selected for the experiments and acclimated to each treatment for 48 hours prior to assay measurements.

The treatments consisted of 6 levels of increasing $pCO_2$, from pre-industrial to expected atmospheric $CO_2$ conditions for the year 2100 in the scenario of business-as-usual (IPCC RCP 8.5: 280, 400, 520, 640, 880, 1120 µatm). $CO_2$ was manipulated by bubbling pre-mixed compressed air from tanks at the desired concentrations into each treatment chamber from the bottom to equilibrate the water to the desired treatment and checking equilibrium by measuring constant pH and DO with a hand-held sensor (HQ40D, Hach Lange). For each species, 4 to 6 replicate polycarbonate chambers (11.5 cm diam, 1L), filled with sand-filtered seawater from the Bigelow system, were used per treatment. A single intact macroalgal thallus or seagrass shoot was attached to the bottom of each chamber from the holdfast or rhizome respectively. The macrophyte/seawater volume ratio in the chambers were similar across species; control chambers with no macrophytes were also included (124 chambers in total). During acclimation, chambers were maintained at ambient temperature inside a recirculating water bath and irradiance was maxed at 260 µmol photons $m^{-2}$ $s^{-1}$ of photosynthetic active radiation on a 12:12 h light:dark cycle. This irradiance level is simulating natural light conditions measured at the collection sites 2 m below surface (S1 Fig in S1 File), and represents saturating light levels for photosynthesis in all of the marine macrophyte species considered in these experiments [45–48]. After acclimation, chambers were sealed, and 1.5 h incubations were conducted at maximum irradiance, in which ambient temperatures varied from 14 to 16˚C (S1 Table in S1 File). Initial and final measurements of seawater carbonate chemistry parameters (pH and total alkalinity, see details below) and dissolved oxygen were recorded for each replicate. During incubations, chambers were mixed by magnetic stir bars. The duration of the incubations was determined during pre-trials where pH change in the chambers was measured at different time intervals (0, 30, 60 and 90 min) confirming that metabolic rates showed a linear response (i.e., constant change in pH) through time. After incubations, all macrophytes were dried at 60 ˚C until constant mass was obtained (~48 h).

**Experiment 2 –Residence time and irradiance.** To examine the effects of water residence time upon *S. latissima* capacity to ameliorate seawater acidity under ambient and simulated future scenarios of climate change in a gradient of irradiance, an experiment was conducted indoors using tanks with a flow-through seawater system based at the Bigelow facilities in November 2018 (S2 Fig in S1 File). Juvenile thalli of *S. latissima* free from epiphytes were collected from the holdfast at the Bigelow dock in the Damariscotta River Estuary Region (43˚ 51'38.0"N, 69˚34'41.7"W). The treatments consisted of 2 levels of environmental conditions (ambient, 400–600 μatm $pCO_2$ and 11 ˚C, and future, 1500–1700 μatm $pCO_2$ and 13 ˚C) and 2 levels of residence time determined by water volume flow (low flow, 0.5 L min$^{-1}$ equivalent to 60 min residence time and high flow, 1.4 L min$^{-1}$ equivalent to 21 min of residence time). These values were chosen based on measurements during changing tides in a *S. latissima* seaweed farm nearby collection sites, and for being acceptable values for cultivating shellfish [49]. In the tanks, those values are equivalent to 0.02 cm s$^{-1}$ and 0.06 cm s$^{-1}$ of linear flow, but we chose to use the units L min$^{-1}$ as it allows for comparisons based on the residence time of a particular volume of water. These conditions were kept constant through the course of the experiment (S2 Table in S1 File). During the experiment, photosynthesis-irradiance curves were generated by subjecting the aquaria to increasing levels of irradiance (n = 6 levels; 0%, 10%, 33%, 60%, 80%, 100% relative to highest photosynthetic active radiation, ~260 μmol photons m$^{-2}$ s$^{-1}$, measured at the collection site, S1 Fig in S1 File).

Temperature and $CO_2$ were manipulated in two mixing tanks, one representing ambient conditions, where no manipulation was done to seawater, and the other future conditions, where $CO_2$ was bubbled into seawater using solenoid valves and a Neptune Apex Controller pH system (feedback sensors were in the mixing tanks; resolution <200 μatm $pCO_2$) and temperature was manipulated using an in-line heater (S2 Fig in S1 File). Water volume flow was controlled by setting ball valves in header tanks and measuring outflow rates with a flowmeter (Blue-White Industries F-440). For each treatment five replicate flow-through culturing tanks with transparent walls were used (38.1 cm * 38.1 cm * 22.8 cm, ~30L). Similar kelp biomass was added to each tank (~150 g fresh weight, 3–5 large thallus) and the biomass/seawater volume ratio in the tanks were similar across treatments. An extra tank was used as a control with no kelp per each treatment (24 tanks in total). All samples were allowed to acclimatize to the treatments in flow through water for 48 h prior to assays, during which the chambers were kept at 260 μmol photons m$^{-2}$ s$^{-1}$ on a 12:12 h light: dark cycle (simulating conditions at the collection site, S1 Fig in S1 File). All tanks were randomly distributed in two rows, where running seawater was supplied to each tank from the header tanks, half of the tanks for the ambient conditions, and the other half for the simulated future conditions. Seawater was supplied into the top end of the rectangular tank and drained through the bottom, opposite end following the longest axis of the tank, where water flow had a constant direction and no obvious stagnant regions were present; each tank was filled to capacity and had an airtight lid fitted with a gasket seal so no head space was present. LED lights (Aquaillumination Hydra 64) were mounted above the tanks illuminating the entire tank surface area homogeneously. Light irradiance levels and spectra were controlled digitally via the LED control panel, starting with dark and increasing light at each time step. Each of the time steps was typically 1.5 h long. Chlorophyll A in seawater content measured at each time step in control tanks was between 0 and 3 μg L$^{-1}$, indicating that most productivity observed during the experiment could be associated to the kelp. After incubations, kelp was dried at 60 ˚C until constant mass was obtained (~ 48 h).

## Monitoring of seawater carbonate system and dissolved oxygen

For the comparison among species (Experiment 1), monitoring of seawater pH, alkalinity, oxygen, temperature and salinity conditions was conducted on the chambers *before* and *after* the incubations. Values of temperature, salinity, oxygen and pH were recorded in all the chambers using a multiparameter meter (HQ40D, Hach Lange) calibrated before measurements with TRIS buffer, from A. Dickson laboratory, with an accuracy of 0.01 pH units. Discrete water samples were acquired from three representative chambers from each $CO_2$ level per species for analysis of total alkalinity ($A_T$) as minimal variation was found among chambers (<10 μmol kg$^{-1}$) thus not reflecting changes among them. These values were averaged to determine initial and final $A_T$ per species for further analysis. Seawater $A_T$ was measured using open-cell titration with triplicate samples (Metrohm 855 Robotic Titrosampler, Metrohm, USA) using 0.1 N HCl (Fisher Chemical) diluted to a nominal concentration of 0.0125 M. Acid was calibrated by analyzing Certified Reference Material (CRM Batch 138) from A. Dickson's laboratory [50]. For each set of triplicate analyses, the median value was considered. Instrumental precision was within 5 μmol kg$^{-1}$.

A similar monitoring was conducted in the flow-through tanks (Experiment 2), where all measurements were done simultaneously on the *inflow* of seawater and the *outflow* of each tank after being exposed to each light step for at least 60 min. This time period was decided based on high frequency continuous oxygen measurements made in one tank per treatment, where a 60-minute wait was enough to see stable changes in productivity (S3 Fig in S1 File). Temperature, salinity and oxygen were measured using a multiparameter meter (HQ40D, Hach Lange). Discrete water samples were acquired from each tank for analysis of pH and total alkalinity ($A_T$) that were run the same day of the experiment. Seawater pH of the discrete water samples was determined on the total scale as per Dickson *et al*. 2007 [51] with a high-precision spectrophotometer (Agilent Cary 8454 with PCB 1500 Water Peliter System) using unpurified m-cresol purple dye (Alfa Aesar A180025-06) [52]. Water samples were kept in a temperature-controlled water bath at 20°C before analysis, and during analysis using a thermostatic cell holder, to minimize temperature-induced errors in absorbance measurements. The spectrophotometer was validated by analyzing TRIS buffer revealing that the system was accurate to within 0.005 pH units. Seawater $A_T$ was measured using a CONTROS HydroFIA TA® instrument (4H-Jena GmbH, Jena Germany), which performed a single-point titration of seawater with 0.1N HCl, using bromocresol green as the indicator for spectrophotometric pH detection, a technique developed by Yao and Byrne (1998) [53] and refined by Li *et al*. (2013) [54]. HCl titrant was calibrated with Certified Reference Material (Batch 172) from A. Dickson's laboratory [50]. Instrumental precision was within 5 μmol kg$^{-1}$.

## Data analysis and statistics

DIC and Ω, as aragonite saturation state, were calculated from pH and $A_T$ (S1 and S2 Tables in S1 File). Carbonate system calculations were performed using the *seacarb* R package [55] and assuming published values for constants $K_1$ and $K_2$ [56], $K_F$ [57], and $K_S$ [58]. Uncertainties of the derived parameters (DIC, $pCO_2$, and Ω) were quantified using a Monte Carlo analysis (100 simulations; Takeshita et al. 2021) [59]. For each simulation, normally distributed errors were introduced into pH (± 0.01) and $A_T$ (± 5 μmol kg$^{-1}$). The overall uncertainties of the derived parameters were calculated as one standard deviation of the simulations. On average, uncertainties for DIC were 0.28%, for $pCO_2$ were 2.5% and for Ω 0.01%. For pCO2, uncertainty is higher at lower $pCO_2$, whereas uncertainty is lower for Ω at lower Ω. The uncertainty for DIC was similar to the uncertainty in $A_T$, that is ± 5 μmol kg$^{-1}$.

To check for differences among species on the effects of $pCO_2$ in the release of DO, uptake of DIC and change in pH and $\Omega$ during incubations (Experiment 1), we used linear models with "species" as categorical variable (4 levels) and "$pCO_2$ level" (initial $pCO_2$ levels before incubations) as a continuous variable including an interaction term between them. Linear models were developed with the lm() function from "base" R package. The release of DO, uptake of DIC and change in pH and $\Omega$ during incubations were calculated as the difference between final and initial levels of DO, DIC and $\Omega$ in the chambers (i.e., rate of change to values before incubations). In order to standardize the comparison among species, the different response variables were normalized by dividing per dry weight biomass of marine macrophytes.

To check for differences among water residence time levels on the release of DO as a function of irradiance in ambient and simulated future scenarios (Experiment 2), we first fitted the photosynthesis irradiance curves, (hereinafter light curves), and then compared the curved derived parameters ($P_{max}$, $E_k$ and $\alpha$) as described below. As there were no signs of photoinhibition, curves were fitted to the model of Jassby and Platt (1976) [60] using the Nelder-Mead method, fitJP() function from "phytotools" R package [61]. The model describes the linear increase in photosynthetic rates with irradiance, up until the saturating irradiance where photosynthesis plateaus at the maximum rate:

$$P = P_{max}*\tanh(\alpha*E/P_{max}) \tag{1}$$

Where P represents the net photosynthetic rate (measured here as change in dissolved oxygen between tank inflow and outflow in a flow-through system, $\Delta DO$ µmol L$^{-1}$), E represents irradiance (µmol m$^{-2}$ s$^{-1}$), $P_{max}$ ($\Delta DO$ µmol L$^{-1}$) is the maximum photosynthetic rate at saturating irradiance, and $\alpha$ represents photosynthetic efficiency ($\Delta DO$ µmol L$^{-1}$). $E_k$ is the saturation irradiance (µmol m$^{-2}$ s$^{-1}$), and is derived from the model as:

$$E_k = P_{max}/\alpha \tag{2}$$

The Welch two sample t-test was used to compare parameters derived from fit light curves ($P_{max}$, $E_k$ and $\alpha$) between the two residence time levels in ambient and simulated future scenarios, t.test() function from "base" R package. The same light curves approach was followed to check for differences between water residence time levels on the uptake of DIC and change in pH and $\Omega$ as a function of irradiance in ambient and simulated future scenarios. Thus, $\Delta DO$ in P and $P_{max}$ terms can be replaced by $\Delta DIC/\Delta pH/\Delta \Omega$. In the case of DIC, pH and $\Omega$ the aim was to evaluate the effect of water residence time in the macrophytes' amelioration of seawater acidity, while DO was used to evaluate photosynthesis rates.

The release of DO, uptake of DIC and change in pH and $\Omega$, were calculated as the difference between inflow and outflow levels of DO, DIC, pH and $\Omega$ in the tanks (i.e., rate of change in respect to inflow seawater). Data was not normalized per dry weight biomass of marine macrophyte because the same species and biomass was used across treatments in Experiment 2.

Finally, in order to extrapolate the results of Experiment 2 to the ecosystem level, models were developed to predict the release of DO or DIC uptake per biomass of *S. latissima* as a function of water residence time and irradiance in ambient and future scenarios, from which pH and $\Omega$ can be derived. Values of DO release or DIC uptake (P) in a continuous gradient of irradiance (E) were estimated from the average values ($\bar{P}, \bar{\alpha}$) of the parameters derived from the fit light curves of each treatment following Eq 1 (Table 1 and S6 Fig in S1 File), and then

**Table 1. Summary of the statistical analysis applied in the two experiments.**

| Experiment 1- Comparison among species | | | | | Experiment 2- Residence time and irradiance experiment | | | | | | | | | | | | | | | |
| --- | --- | --- | --- | --- | --- | --- | --- | --- | --- | --- | --- | --- | --- | --- | --- | --- | --- | --- | --- | --- |
| | | | | | | AMBIENT | | | | | | | | FUTURE | | | | | | |
| | Df | Sum Sq | F | p-value | | Low Flow | | High Flow | | t | Df | p-value | | Low Flow | | High Flow | | t | Df | p-value |
| | | | | | | Mean | SE | Mean | SE | | | | | Mean | SE | Mean | SE | | | |
| **DO** | | | | | **DO** | | | | | | | | **DO** | | | | | | | |
| Species | 4 | 440893 | 75.6 | **<0.001** | $P_{max}$ | 32.44 | 3.58 | 20.54 | 1.96 | -2.91 | 6.19 | **0.026** | $P_{max}$ | 42.59 | 4.00 | 22.56 | 2.58 | -4.06 | 6.79 | **0.005** |
| pCO$_2$ | 1 | 6134 | 4.2 | **0.043** | $E_k$ | 210.97 | 29.78 | 203.98 | 18.09 | -0.20 | 6.60 | 0.847 | $E_k$ | 182.67 | 9.66 | 152.52 | 7.55 | -2.35 | 7.00 | 0.051 |
| Species*pCO$_2$ | 4 | 9279 | 1.6 | 0.181 | α | 0.16 | 0.02 | 0.10 | 0.01 | -3.39 | 5.78 | **0.015** | α | 0.23 | 0.01 | 0.15 | 0.01 | -4.44 | 6.99 | **0.002** |
| Residuals | 114 | 166184 | | | | | | | | | | | | | | | | | | |
| **DIC** | | | | | **DIC** | | | | | | | | **DIC** | | | | | | | |
| Species | 4 | 1182321 | 56.6 | **<0.001** | $P_{max}$ | 19.15 | 1.45 | 7.32 | 0.01 | -7.29 | 3.00 | **0.005** | $P_{max}$ | 25.79 | 3.69 | 22.89 | 9.28 | -0.22 | 2.61 | 0.830 |
| pCO$_2$ | 1 | 53957 | 10.3 | **0.002** | $E_k$ | 121.89 | 16.76 | 89.03 | 7.45 | -1.48 | 3.98 | 0.212 | $E_k$ | 174.37 | 40.15 | 134.82 | 68.42 | -0.38 | 3.23 | 0.720 |
| Species*pCO$_2$ | 4 | 84011 | 4.0 | **0.004** | α | 0.16 | 0.01 | 0.08 | 0.01 | -4.52 | 3.67 | **0.012** | α | 0.17 | 0.03 | 0.14 | 0.05 | -0.34 | 3.14 | 0.750 |
| Residuals | 114 | 594832 | | | | | | | | | | | | | | | | | | |
| **pH** | | | | | **pH** | | | | | | | | **pH** | | | | | | | |
| Species | 4 | 4 | 48.9 | **<0.001** | $P_{max}$ | 0.05 | 0.00 | 0.02 | 0.00 | -7.63 | 3.00 | **0.005** | $P_{max}$ | 0.13 | 0.02 | 0.11 | 0.08 | -0.75 | 3.89 | 0.490 |
| pCO$_2$ | 1 | 1 | 64.9 | **<0.001** | $E_k$ | 104.93 | 13.21 | 61.77 | 0.00 | -2.92 | 3.00 | 0.006 | $E_k$ | 234.83 | 54.30 | 167.29 | 43.85 | -0.73 | 2.61 | 0.524 |
| Species*pCO$_2$ | 4 | 1 | 16.0 | **<0.001** | α | 0.00 | 0.00 | 0.00 | 0.00 | -4.48 | 3.00 | **0.021** | α | 0.00 | 0.00 | 0.00 | 0.00 | 0.46 | 1.37 | 0.708 |
| Residuals | 114 | 2 | | | | | | | | | | | | | | | | | | |
| **Ω** | | | | | **Ω** | | | | | | | | **Ω** | | | | | | | |
| Species | 4 | 97 | 34.5 | **<0.001** | $P_{max}$ | 0.17 | 0.00 | 0.06 | 0.00 | -7.19 | 3.00 | **0.005** | $P_{max}$ | 0.11 | 0.02 | 0.08 | 0.03 | -0.49 | 2.92 | 0.650 |
| pCO$_2$ | 1 | 3 | 4.8 | **0.031** | $E_k$ | 126.82 | 16.66 | 96.89 | 7.49 | -1.35 | 3.97 | 0.247 | $E_k$ | 146.67 | 56.24 | 108.18 | 37.77 | -0.37 | 1.62 | 0.740 |
| Species*pCO$_2$ | 4 | 11 | 3.9 | **0.006** | α | 0.00 | 0.00 | 0.00 | 0.00 | -5.46 | 3.68 | **0.006** | α | 0.00 | 0.00 | 0.00 | 0.00 | -0.54 | 1.35 | 0.660 |
| Residuals | 114 | 80 | | | | | | | | | | | | | | | | | | |

Bold numerical values are statistically significant. DO, dissolved oxygen; DIC, dissolved inorganic carbon; Ω saturation state of calcium mineral (aragonite). Df, degrees of freedom. Sum Sq, Sum of squares. SE, standard error.

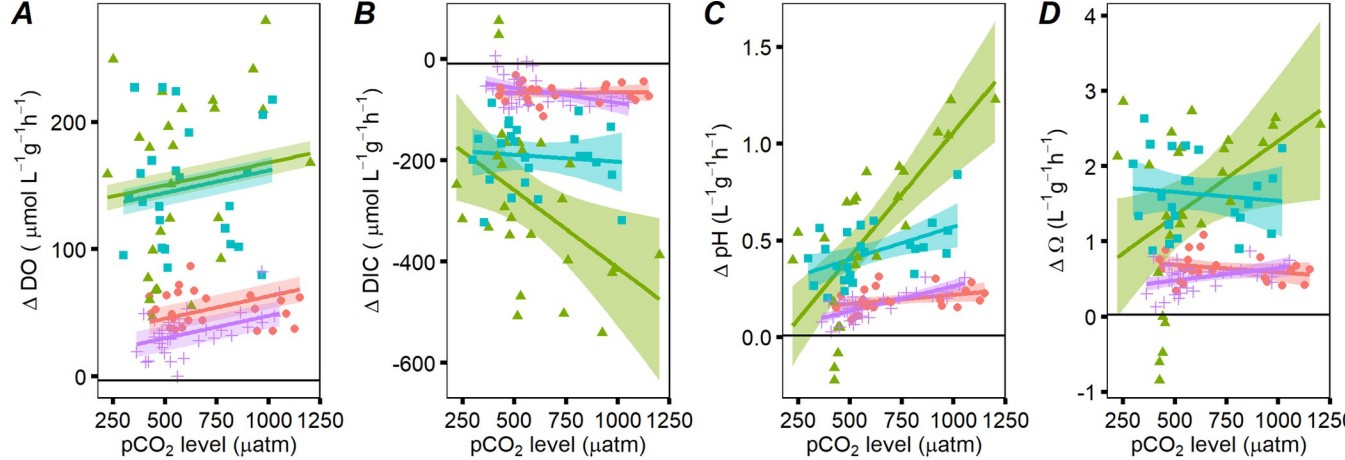

**Fig 1. Capacity to ameliorate seawater acidity derived from CO$_2$ enrichment in four species of marine macrophytes.** Plots represent (**A**) the release of dissolved oxygen (ΔDO), (**B**) the uptake of dissolved inorganic carbon (ΔDIC), (**C**) the change in pH (ΔpH), and (**D**) the change in saturation state of calcium mineral (aragonite) (ΔΩ) during chamber incubations at different initial pCO$_2$ levels. Values reported are normalized per dry weight biomass of macrophyte, volume of chamber and time of incubation. The black horizontal line represents the average value in the control treatments with no macrophytes. Colored lines show predicted average values and shade areas 95% CI in *Saccharina latissima* (green-triangles), *Ulva lactuca* (blue-squares), *Zostera marina* (purple-crosses), and *Fucus vesiculosus* (red-circles).

divided by biomass (B) of macrophyte and volume (v) of the tank.

$$P = \bar{P}_{max} * \tanh(\bar{\alpha} * E / \bar{P}_{max}) / (B^* v) \tag{3}$$

Models were built assuming a linear relationship between water residence time and response variables, and the plots presenting the model predictions were developed with the vis-reg2d() function from the "visreg" R package [62]. Models were developed with biomass of macrophyte expressed as fresh weight. The following conversion, built with data from this study, can be used to calculate model predicted values expressed as dry weight (Dry Weight = 0.1187*Fresh Weight—0.0086, df = 30, p<0.001, $R^2$ = 0.9918).

All data and statistical analysis presented were done in the R software (R 4.2.1) [63]. For all statistical analyses, normality and homoscedasticity were assessed via visual estimation of trends of model residuals and data was not transformed.

## Results

### Experiment 1

All marine macrophyte species showed amelioration of seawater acidity with respect to the control treatments (Fig 1). The overall effect of increased seawater $pCO_2$ and the differences among macrophyte species on the release of DO were significant while the interaction between species and pCO2 level was not, indicating that, on average *S. latissima* and *U. lactuca* showed the higher release of DO compared to *F. vesiculosus* and *Z. marina*, and on average (across species), $pCO_2$ increased the release of DO (Fig 1A and Table 1 and S3 Table in S1 File). The effect of $pCO_2$—and the differences among species—on the uptake of DIC were also significant, as was the interaction (Fig 1B and Table 1 and S3 Table in S1 File). On average, uptake of DIC was larger in *S. latissima* and *U. lactuca*, and on average (across species), higher $pCO_2$ increased the uptake of DIC, although in this case the differences among species were based on the level of $pCO_2$ (significant interaction) where the major response to increased $pCO_2$ was shown by *S. latissima* and *Z. marina* (steeper slopes). Thus, for every 1-microatmosphere increase in $pCO_2$, *S. latissima* exhibited a DIC uptake of an additional 0.30 μmol $L^{-1}$ $g^{-1}$ $h^{-1}$, *Z. marina* 0.05 μmol $L^{-1}$ $g^{-1}$ $h^{-1}$, *U. lactuca* 0.02 μmol $L^{-1}$ $g^{-1}$ $h^{-1}$ while *F. vesiculosus* decreased uptake by 0.01 μmol $L^{-1}$ $g^{-1}$ $h^{-1}$. The effect of $pCO_2$ on the change in pH and Ω was also significant, as were the difference among species and the interaction (Fig 1C and Table 1 and S3 Table in S1 File). Thus, on average *S. latissima* and *U. lactuca* showed a bigger change in pH and Ω, but the responses to $pCO_2$ were larger in *S. latissima* and *Z. marina* (steeper slopes) (Fig 1C and 1D and Table 1 and S3 Table in S1 File). For *S. latissima*, *Z. marina*, *F. vesiculosus* and *U. lactuca* the change in pH increased with the level of $pCO_2$ (positive slope, 0.0013, 0.00026, 0.000093 and 0.00034 change per unit of $pCO_2$, respectively) (Fig 1C). For *S. latissima* and *Z. marina*, the change in Ω increased with the level of $pCO_2$ (positive slope, 0.0031 and 0.0005 change per unit of $pCO_2$, respectively), and for *F. vesiculosus* and *U. lactuca* the change in Ω decreased when the level $pCO_2$ increased (negative slope, -0.00031 and -0.00037 change per unit of $pCO_2$, respectively) (Fig 1D). Values of DO, DIC, pH and Ω of each treatment are presented in S1 Table in S1 File. Speciation of DIC forms can be seen in Suppl (S4 Fig in S1 File). Among all the species analyzed, *S. latissima* had the largest impact on carbonate chemistry, able to remove the most DIC and alter pH and Ω more substantially also when $pCO_2$ increased, making it a promising candidate for acidification mitigation measures under elevated $pCO_2$ conditions.

## Experiment 2

The effect of elevated $pCO_2$ was also seen in this experiment where the influences of water residence time along a gradient of irradiance were assessed. In this experiment, kelp's impact on the release of DO, uptake of DIC and changes in pH and $\Omega$ with respect to the control treatments with no kelps was larger under future than ambient conditions (Fig 2). In the kelp treatments, all response variables measured showed a linear increase (initial slope, $\alpha$) with irradiance before plateauing off ($P_{max}$) above saturating irradiance ($E_k$) (S6 Fig in S1 File), indicating that the effect of irradiance on the rate of change of DO, DIC, pH and $\Omega$ between inflow and outflow increased (as expected) with the light. For DO (Fig 2A and 2B and S6 Fig in S1 File): $P_{max}$ was significantly higher at the high residence time level, $E_k$ presented no differences, and $\alpha$ was significantly higher at high residence time in both ambient and future conditions (Table 1). For DIC, pH and $\Omega$ (Fig 2C to 2H and S6 Fig in S1 File): $P_{max}$ was significantly higher at high residence time, $E_k$ presented no differences, and $\alpha$ was significantly higher at high residence time in ambient scenarios (Table 1). No differences between low and high residence time were found for any of the light curve parameters in future scenarios (Table 1). Values of DO, DIC and $\Omega$ of each treatment are presented in S2 Table in S1 File. Speciation of DIC forms can be seen in the Supplementary (S5 Fig in S1 File).

## Model

Predicted levels of change in DO and DIC per gram of *S. latissima* (sugar kelp) showed a general decrease in the irradiance threshold for OA amelioration from ambient to simulated future scenarios, with important effects of water residence time (Fig 3, where the black line indicates the threshold for amelioration). This means that, above that threshold, for the same level of irradiance, the amount of DIC sequestered, DO released, or change in pH and $\Omega$ in future scenarios will be larger compared to ambient scenarios. The model is expressed per gram of sugar kelp in fresh weight per liter of water. We provided the conversion to dry weight to facilitate further calculations at the ecosystem scale (see methods). If the daily light cycle profile is known cumulative or daily changes can be calculated.

## Discussion

Results of this study indicate that while marine macrophyte species increased their productivity under experimentally elevated $pCO_2$, certain species had a greater amelioration capacity of seawater acidity (ability to alter the surrounding seawater carbonate system) than others. Of all the species analyzed, *S. latissima* (sugar kelp) stands as the most promising candidate, able to remove the most DIC and alter pH and $\Omega$ more substantially, effectively counteracting acidification as $pCO_2$ increases. Furthermore, the amelioration capacity of sugar kelp was optimized in conditions of high irradiance and high residence time. However, the effect varied between ambient and simulated future environmental conditions, where the differences observed between residence time treatments were more variable and less significant under future conditions. Overall, this research provides evidence of the variability existent among marine macrophytes in their ability to alter the seawater carbonate system and quantifies the effects of the different drivers.

Under equal conditions, species-specific inorganic carbon physiology can determine the differences in the amelioration of seawater acidity among marine macrophytes species. This is particularly clear under the studied present-day conditions, where *S. latissima* and *U. lactuca* showed major effects on the seawater carbonate chemistry, likely reflecting a higher efficiency of their carbon concentration mechanisms to utilize $HCO_3^-$ compared to *Z. marina* [64] and *F. vesiculosus* [46]. However, with the increase of $CO_2$, *S. latissima* and *Z. marina* increased

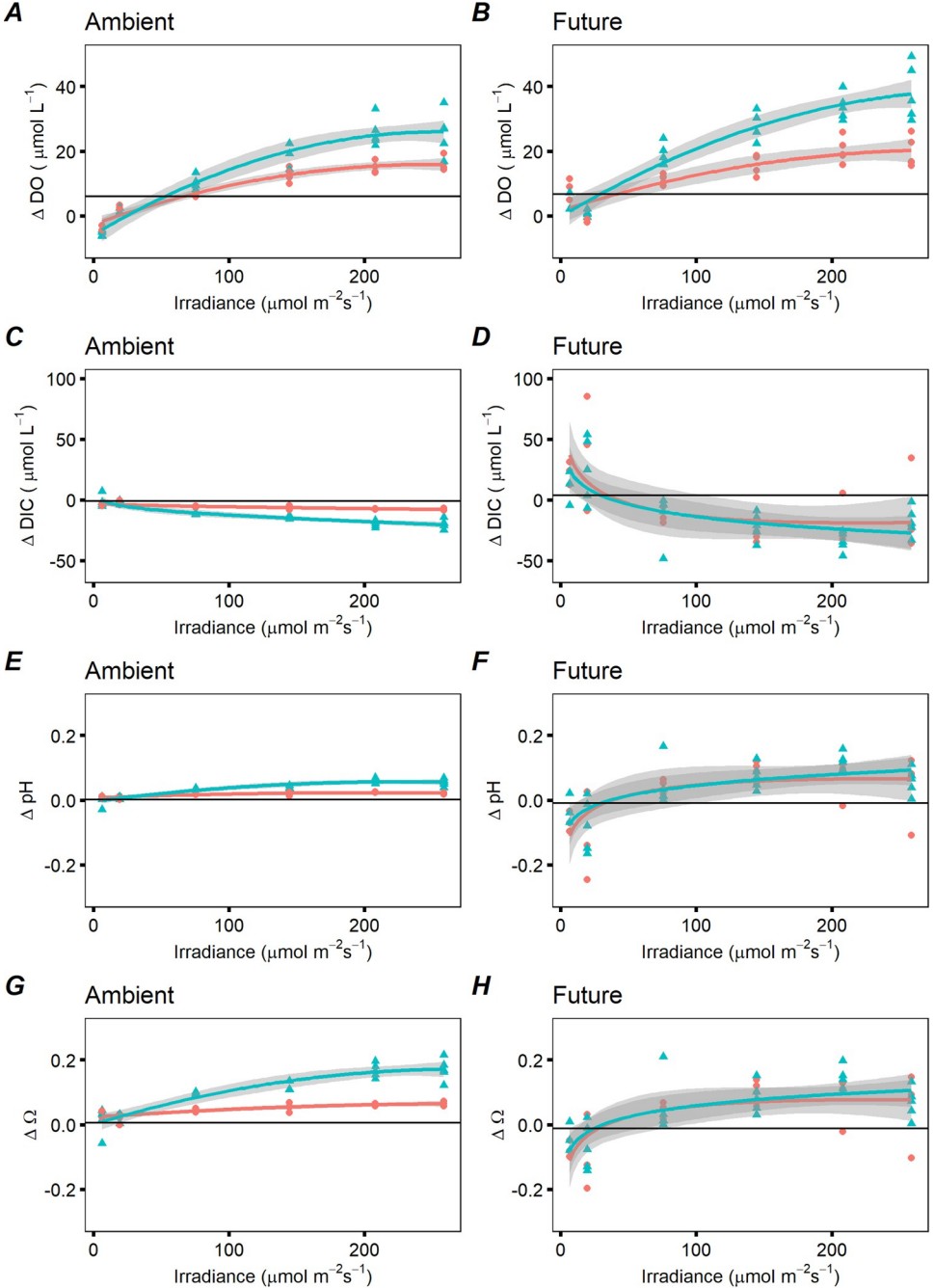

**Fig 2. Capacity to ameliorate seawater acidity in *Saccharina latissima* as a function of light and water residence time.** Plots represent (**A, B**) the release of dissolved oxygen (ΔDO), (**C, D**) uptake of dissolved inorganic carbon (ΔDIC), (**E, F**) the change in pH (ΔpH) in the tanks with sugar kelp, and (**G, H**) the change in saturation state of calcium mineral (aragonite) (ΔΩ) in the tanks with sugar kelp. The black horizontal line represents the average value in the control treatments with no kelp. Colored lines show a smoothing with the loess method for values in low- (blue-triangles) and high- water flow (red-circles) and grey areas 95% CI.

primary productivity, and overall impact upon buffering seawater (normalized to dry weight), more strongly than *U. lactuca* and *F. vesiculosus*, which is probably related to the saturation state of $CO_2$, or $CO_2$ limitation, for each species [65]; although is important to note that

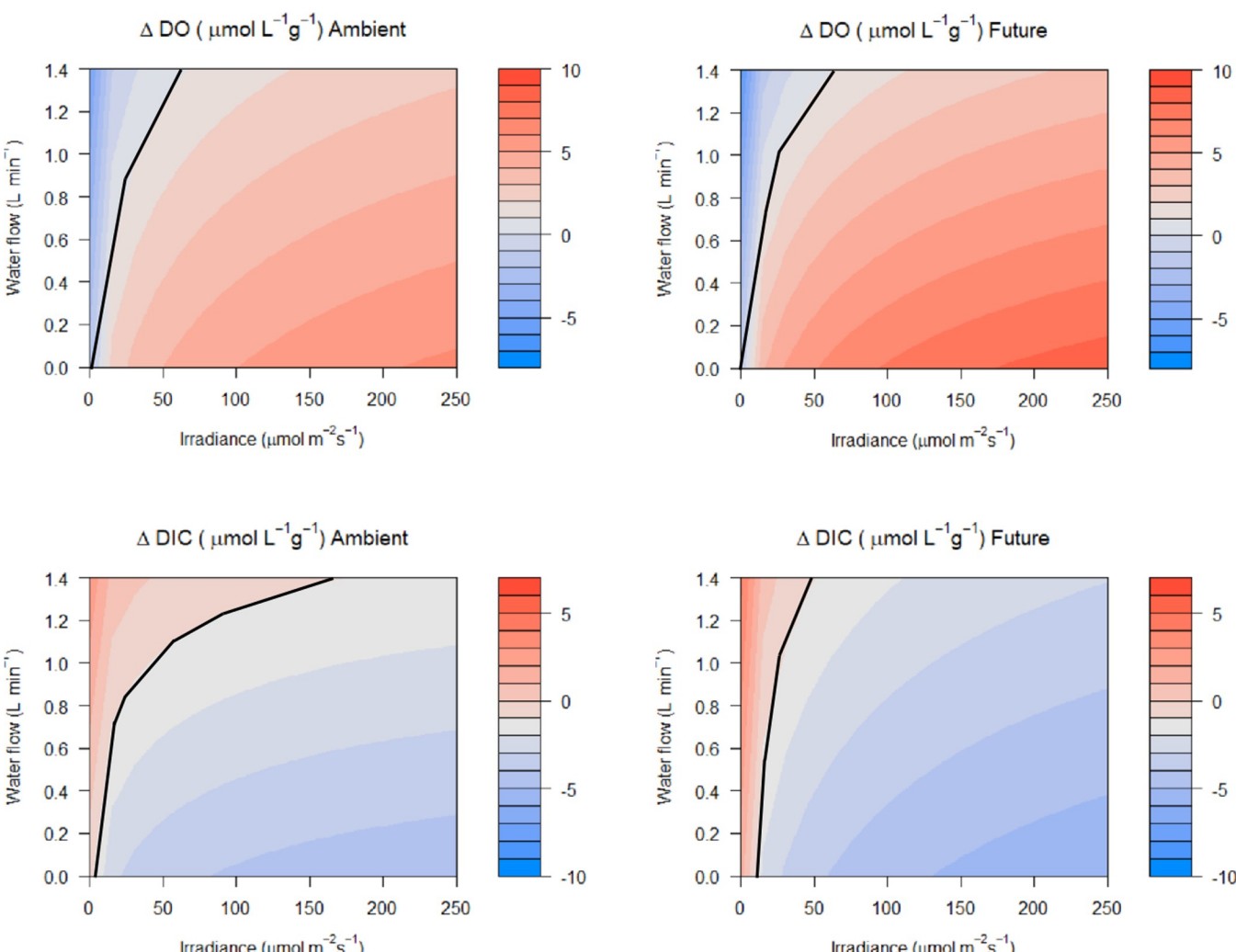

**Fig 3. Predictive model results for *Saccharina latissima*.** Model results of how much change to expect in dissolved oxygen (ΔDO) and dissolved inorganic carbon (ΔDIC) per gram of kelp (Fresh weight) per liter of water depending on water residence time (here shown as flow in L min$^{-1}$) and irradiance in current (ambient) and future (high-$CO_2$ and high-temperature) environmental conditions. Black line marks the threshold for the amelioration effect. Values of DO and DIC release or uptake in a continuous gradient of irradiance were estimated from the average parameters derived from the light curves. The model is based in a linear interaction between water residence time and the change in DO or DIC. Units of change (μmol L$^{-1}$ g$^{-1}$) are shown in the legend. By knowing the change in DIC, the change in pH and Ω can be calculated (see discussion).

seagrass belowground tissue is usually buried in the sediments and having it exposed in the water column as in this study could increase respiration effects in seawater chemistry that should be further studied. All the species studied here are capable of using both $HCO_3^-$ and $CO_2$ through carbon concentration mechanisms, and all of them apparently use $CO_2$ to a greater extent as concentrations increase [24, 64, 65]. However, *Z. marina* and *S. latissima* are not saturated by $CO_2$ in present-day or future conditions (at least for the future conditions studied here, see slopes in Fig 1B), while *U. lactuca* and *F. vesiculosus* are [65]. Therefore, *Z. marina* and *S. latissima* may be capable of utilizing additional $CO_2$ by diffusion if external seawater $CO_2$ concentrations exceed those produced internally by the carbon concentration mechanisms. It follows that, if a macrophyte species is currently limited in $CO_2$, and uses carbon concentration mechanisms, an increase in $CO_2$ availability will translate into more productivity, growth and/or higher photosynthetic rates [22, 24, 66]. The amelioration effect in

seawater acidity will then be higher, as observed in this study for *S. latissima*, which can release DO, remove DIC, and increase pH and $\Omega$ in seawater relatively more than the other species studied.

Our results also suggest that higher availability of $CO_2$, combined with slightly higher temperature (in our case within the range of optimum growth of the species studied), enhances the effect of light on sugar kelp, since for the same irradiance in simulated future conditions compared to present-day conditions, the seawater levels of DO increase and levels of DIC decrease. This supports the supposition that productivity (i.e., $P_{max}$) of photosynthesis in marine macrophytes will increase in the future [67, 68]. Although light is the primary source of photosynthetic energy, the amelioration capacity can also be controlled by the combined effects of the concentration of DIC forms in the bulk fluid [17] and the influence of water flow on DIC and DO transport to and from the blade surface through the diffusive boundary layer [28], as well as nutrient levels [69]. Thus, the fact that we observed differences between residence time levels in ambient conditions in the uptake of DIC and change in $\Omega$, but not in simulated future conditions, might be related with more $CO_2$ available for photosynthesis in a high $CO_2$ setting. If an unlimited amount of $CO_2$ is available (i.e., saturation) to the macrophyte, residence time will neither impact its flux, nor the acidification amelioration capacity. Yet, the slight increase in temperature in the future conditions' treatment can decrease solubility of $CO_2$ as well as increase the macrophyte metabolic activity which can limit its availability. Another factor to consider is that we derived DIC from pH and alkalinity measurements, a calculation which could propagate errors [70] that enlarge with effect size, leading to reduced power to discern differences among treatments for carbonate chemistry parameters in the simulated future conditions. This complication is not so for DO, which was measured independently. The small but significant relative increase in DO in simulated future conditions for each respective residence time or water volume flow rate suggests an increase in photosynthetic rates, but more direct measurements via Pulse Amplitude Modulated (PAM) fluorometry are required to discern definitively. Overall, the capacity of sugar kelp to ameliorate seawater acidity will likely increase in future conditions due to higher availability of $CO_2$, but the effects of water residence time are not apparent.

The physiological mechanisms underpinning the responses of marine macrophytes to increased $CO_2$, and how $CO_2$ levels influence the methods of DIC uptake, are poorly understood [24]. If certain macrophytes are capable of utilizing additional $CO_2$, a shift in the relative proportion and concentration of each DIC species expected in future conditions may lead to larger amelioration capacity of some macrophytes over others. However, the different environmental conditions, such as light availability or those that could affect water residence time, might vary, or even offset their potential to ameliorate the acidity of seawater. Environmental conditions vary at multiple spatial and temporal scales, and whether the amelioration capacity of marine macrophytes is consistent and strong enough in space and time to confer remediation for organisms susceptible to ocean acidification and future environmental change is still a matter of concern. For instance, in present-day conditions, sugar kelp in a polar region has been shown to be $CO_2$ saturated [71], indicating that geographical location and likely temperature can play an important role on the final amelioration effect independently of the way of acquiring $CO_2$. In addition, several studies have quantified the temporal variability of the seawater carbonate system within different marine macrophyte systems, showing high frequency fluctuations mostly related with daily light cycles and flow [9, 18, 19, 30, 72]. Therefore, the suitability and scenarios under which marine macrophytes can be used as a mitigation tool remains uncertain.

We have found experimental evidence that, if residence time and light conditions are optimal, some marine macrophytes can sequester DIC and raise calcium carbonate saturation

state ($\Omega$), likely improving conditions for calcification [73] and mitigating episodes of extreme low pH (i.e. upwelling of corrosive waters or episodic terrestrial freshwater inputs). However, these changes might be different during night time, as we have found hints that no-light (i.e. dark) respiration in our experiments could increase acidification in future conditions accordingly to what has been proposed by [74]. Here we provide a framework that, for a known water volume flow or residence time, irradiance (as photosynthetic active radiation), and standing stock of sugar kelp biomass, can predict the magnitude of the current and future capacity to improve seawater conditions for subsequent ocean acidification mitigation purposes. We apply our model to a specific body of water in temperate regions within the boundaries of the macrophyte ecosystem. We make assumptions such as all kelp will be subjected to the same light levels and are free from epibionts, although in more realistic scenarios, self-shading, light downwelling and biofouling could occur. Despite this, rough calculations can be made for the ideal scenario. For instance, in future conditions, at water flow of 0.5 L min$^{-1}$ and 250 µmol photons m$^{-2}$ s$^{-1}$ of photosynthetic active radiation (see Fig 3, $\Delta$ DIC Future -4.5 µmol L$^{-1}$ g$^{-1}$) in a natural seaweed bed patch or around an aquaculture line of sugar kelp of 5,000 kg FW in 1000 m$^3$ of water, we can predict a change in DIC of 22.5 µmol L$^{-1}$. If information on other carbonate system parameters (at least one) is available for that body of water, inferences can be made on how much change to expect in pH or $\Omega$ [51]. In the example above, assuming initial values of DIC = 2200 µmol kg$^{-1}$, 1 L$_{seawater}$ = 1.024 kg, total alkalinity = 2100 µmol kg$^{-1}$, salinity = 30 and temperature = 11 C, the pH (total scale) will change from 7.15 outside to 7.27 within the sugar kelp habitat. Linear relation among biomass and metabolically linked seawater chemistry responses has been demonstrated in [14, 75] for other marine macrophyte species, but not yet for sugar kelp.

The facilitative interaction between marine macrophytes and organisms vulnerable to ocean acidification, such as calcifying organisms, has been confirmed in several studies [14, 15, 75–78]. In fact, co-culture with marine macrophytes is being proposed as a mitigation tool in shellfish aquaculture impacted by ocean acidification [79–83], and will also counteract deoxygenation of seawater [81]. However, as demonstrated in this study when or where amelioration of ocean acidification by marine macrophytes will consistently translate into favorable conditions for other organisms will depend on the species' physiology and also on factors such as light, residence time, and $CO_2$ enrichment, studied here; each should be considered for management actions.

In conclusion, our results in experimentally elevated $CO_2$ conditions demonstrate that different species of marine macrophytes show different potential to ameliorate seawater acidity, where $CO_2$ enrichment, residence time and light availability have a critical effect. Sugar kelp, *S. latissima*, stands alone in its capacity to alter the seawater carbonate system among the species studied, and its amelioration potential increases in high $CO_2$ settings, although the effect of residence time only appears to be relevant in present-day ambient conditions. This study supports the hypothesis that marine macrophytes can locally alleviate consequences of human-induced ocean acidification. Nevertheless, these findings are limited to a laboratory setting in the absence of air-water exchange and more studies in the field are crucial to understand the biological and physical drivers of coastal $CO_2$ stoichiometry.

## Supporting information

**S1 File.**
(PDF)

**S1 Data.**
(CSV)

**S2 Data.**
(CSV)

**S3 Data.**
(CSV)

**S4 Data.**
(CSV)

## Acknowledgments

We thank Emily Donham, Colleen George, Hilary Neckles, Mike Doan and Melisa Meléndez-Oyola for their help during collections in the field and/or laboratory analyses.

## Author Contributions

**Conceptualization:** Aurora M. Ricart, Joseph Salisbury, Suzanne N. Arnold, Nichole N. Price.

**Data curation:** Aurora M. Ricart, Brittney Honisch, Evangeline Fachon, Christopher W. Hunt.

**Formal analysis:** Aurora M. Ricart, Brittney Honisch, Evangeline Fachon.

**Funding acquisition:** Joseph Salisbury, Nichole N. Price.

**Investigation:** Aurora M. Ricart, Nichole N. Price.

**Methodology:** Aurora M. Ricart, Brittney Honisch, Evangeline Fachon, Christopher W. Hunt, Suzanne N. Arnold, Nichole N. Price.

**Supervision:** Brittney Honisch, Nichole N. Price.

**Writing – original draft:** Aurora M. Ricart.

**Writing – review & editing:** Brittney Honisch, Evangeline Fachon, Christopher W. Hunt, Joseph Salisbury, Suzanne N. Arnold, Nichole N. Price.

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
