## [Decision Letter · Decision Letter 0]

18 Jan 2023

PONE-D-22-32214Optimizing marine macrophyte capacity to locally ameliorate ocean acidification under variable light and flow regimes: Insights from an experimental approachPLOS ONE

Dear Dr. Ricart,

Thank you for submitting your manuscript to PLOS ONE. After careful consideration, we feel that it has merit but does not fully meet PLOS ONE’s publication criteria as it currently stands. Therefore, we invite you to submit a revised version of the manuscript that addresses the points raised during the review process.

Both the reviewers and I think the manuscript presents novel and important data that will further our understanding of macrophyte impacts on carbonate chemistry, and we are willing to consider a revised version for publication in the journal.

Both reviewers brought up some important considerations especially about the impacts of temperature vs CO_2 _concentration as well as statistical treatments. In addition to these I thought there were two important points that should be discussed. I think that the experiments were done without sediment. It might be important to discuss the potential implications of including below ground tissue in the experiments but having that tissue in the water column. Further, I think (but am not certain for experiment 2) all incubations were conducted in the light (when macrophytes might ameliorate acidication). I think it would be important to discuss that we aren’t sure about changes in night time respiration under each of these conditions.

Once again, thank you for submitting your manuscript to PLoS One  and I look forward to receiving your revision.

We look forward to receiving your revised manuscript.

Kind regards,

Laura Reynolds

Academic Editor

PLOS ONE

Journal Requirements:

"This study was supported by NASA (https://www.nasa.gov/) grant NX14AL84G to JS, NOAA (https://www.noaa.gov/) grants N17OAR0170164 to JS & NA17NMF4270202 to NP, the Broad Reach Foundation to NP (https://www.broadreachfoundation.org/), the Nature Conservancy to NP (https://www.nature.org/), and the NSF REU Program to NP (https://www.nsf.gov/crssprgm/reu/) (grants 1156740 and 1460861)."

Reviewers' comments:

Reviewer's Responses to Questions

**Comments to the Author**

1. Is the manuscript technically sound, and do the data support the conclusions?

Reviewer #1: Yes

Reviewer #2: Yes

2. Has the statistical analysis been performed appropriately and rigorously? 

Reviewer #1: Yes

Reviewer #2: Yes

3. Have the authors made all data underlying the findings in their manuscript fully available?

Reviewer #1: Yes

Reviewer #2: Yes

4. Is the manuscript presented in an intelligible fashion and written in standard English?

Reviewer #1: Yes

Reviewer #2: Yes

5. Review Comments to the Author

Reviewer #1: The manuscript Optimizing marine macrophyte capacity to locally ameliorate ocean acidification under variable light and flow regimes: Insights from an experimental approach assessed biotic (i.e., species identity) and abiotic drivers (i.e., light level, residence time) of macrophytes’ ability to ameliorate seawater acidity under ambient and increased pCO2 conditions. The study had three main goals: 1) to compare the ability of four macrophyte species to ameliorate seawater acidity under a range of pCO2 conditions (Experiment 1), 2) to assess the effects of light level and residence time on the best-performing species’ ability to ameliorate seawater acidity under ambient and future pCO2 and temperature conditions (Experiment 2), and 3) to create a model predicting amelioration effects as a function of light level and flow rate.

In Experiment 1, the authors identified Saccharina latissima as the best-performing species in its ability to ameliorate seawater acidity (i.e., consume dissolved inorganic carbon (DIC), increase pH, and increase aragonite saturation state) under increased pCO2 conditions. In Experiment 2, the authors found that S. latissima amelioration effects (relative to control treatments without S. latissima) were greater under future compared to ambient pCO2 and temperature conditions. They also found stronger amelioration effects when residence time was high (or flow rate was low) under ambient conditions, but residence time had no effect under future conditions. The authors interpret this as evidence that as macrophytes become pCO2-saturated, residence time may no longer be a relevant driver of macrophyte seawater acidity amelioration. The predictive model was parameterized by Experiment 2 and predicted amelioration effects via DIC uptake by S. latissima as a function of light level and flow rate.

This work advances discussion on the role of macrophytes in mitigating ocean acidification by evaluating species differences and effects of light and residence time on seawater acidity amelioration in tightly controlled laboratory experiments. Overall, the manuscript is well written, and the study objectives are clear. However, there are some aspects of the experimental design and methodology that should be clarified, and the limitations to interpretation and model extrapolation of results obtained from the laboratory-based studies warrant further discussion.

Major comments:

1. In Experiment 1, I am not sure if an averaged value for total alkalinity for each species and pCO2 level treatment combination is appropriate to use in estimating other carbonate chemistry parameters used as response variables (line 223). While total alkalinity did not vary much between replicates, using an average value for a subset of replicates within the same treatment combination ‘hides’ within-group variability and may be a form of pseudoreplication. If this cannot be properly justified, it may be more appropriate to run the statistical analyses with the three replicates per treatment combination where total alkalinity was actually measured.

2. In Experiment 2, the future conditions included increased temperature by 2°C but there is little attention given to the effects of warming throughout the manuscript. It would be helpful to provide context on whether this is a modest increase in temperature in terms of S. latissima physiology and whether 13°C falls within its tolerance range.

3. There is also no mention of whether the elevated temperature in the future conditions in Experiment 2 could have confounded the effects of residence time and pCO2 levels. The treatments were not independent (i.e., there were no treatments with ambient pCO2 + future temperature or treatments with future pCO2 + ambient temperature), so while the ambient and future conditions may reflect the most realistic combinations of pCO2 levels and temperatures, it’s not possible to rule out a temperature effect here.

4. The predictive model is a bit unclear, so it may be helpful to provide a written formulation of the model. In some parts of the manuscript, the model seems to include biomass as a predictor variable (line 39) but in others, the response variable appears to be standardized by biomass (lines 294-295). If the response variable is change in DIC per gram of biomass, then biomass is not really a predictor in the model but rather the unit of measurement for the response variable. Scaling the response variable by biomass relies on the assumption that metabolism (e.g., DIC uptake) scales linearly with biomass.

5. There could be more discussion on the ecosystem-level implications of the laboratory-based findings. In natural settings, seawater acidity amelioration by macrophytes likely varies over space and time. The authors touched on how basin geomorphology, diurnal cycles, and water clarity will affect the ability of S. latissima to ameliorate seawater acidity. If S. latissima produces the greatest amelioration effects under saturating light conditions compared to the other species, are these effects countered by stronger respiration effects, potentially increasing seawater acidity, under dark conditions? If so, I think it is important to caveat that S. latissima may not be the ideal candidate for acidification mitigation. Additionally, other processes occurring in the water column or benthos (e.g., alkalinity production/consumption related to respiration, air-water CO2 exchange, etc.) can affect acidity. How might these uncertainties affect the reliability of the predictive model?

line 106: it may be helpful to define residence time.

lines 130-132: alternation of generations is not particularly useful to include here since it is common in most plants and algae to some degree.

lines 149-150: how did you ensure equilibrium was reached before starting the experiment?

lines 153-154: what was the flow rate?

lines 164-165: how did temperature variation throughout the incubation affect carbonate chemistry calculations?

lines 166: maybe state upfront the measured carbonate chemistry parameters.

line 179: maybe explain that the pH controller system could maintain a pCO2 level within a range of 200 μatm.

lines 199-200: how did you ensure equilibrium was reached before starting the experiment?

lines 284-286: it’s not clear how the same light curves approach was applied to the other response variables. In equations (1) and (2), were P and Pmax terms simply replaced by net ΔDIC/ΔpH/ΔΩ between tank inflow and outflow and the maximum ΔDIC/ΔpH/ΔΩ at saturating irradiance, respectively? I suggest writing out the equations in a more generalized form for clarity. For example, P and Pmax could be replaced with other terms, such as R and Rmax, and state that R is ΔDO/ΔDIC/ΔpH/ΔΩ.

line 292: was leaf surface area accounted for as well?

line 305: it’s best practice to state R version used for analyses.

lines 328-330: should the values mentioned here match the slopes presented in Table S3? If not, where do these numbers come from?

lines 334-339: should the values mentioned here match the slopes presented in Table S3? If not, where do these numbers come from?

lines 357-359: was the higher impact of kelp in future compared ambient conditions assessed qualitatively?

lines 446-447: since only two flow rates (0.02 and 0.06 cm s-1) were used to parameterize the model, are there limitations when interpreting extrapolated flows an order of magnitude higher?

line 469: in the example starting on line 469, the predictive model is applied to an example. The spatial extent of the effect is noted (1000 m3), but what about the temporal extent, i.e., how long would the effect be sustained?

lines 496-503: mention that findings are limited to macrophyte seawater acidity amelioration during photosynthesis in a laboratory setting, in the absence of air-water exchange, etc.

Table 1: label experiments numerically, e.g., ‘Experiment 1 – Comparison among species’ and include statistical analyses used, either in the table or table caption. If ANOVA was used to compare overall effects of species, pCO2, and their interaction for Experiment 1, this should be listed in the methods, too.

Table S1: since the number of replicates per treatment combination (n) varied, consider including n for each treatment combination in this table.

Table S3: how was the linear model constructed so that coefficients were available for each species? I understand that an intercept-free model in lm() can accomplish this, but an intercept is reported in this table. More description in the methods would make the data analyses more reproducible.

Reviewer #2: In this study the capacity of submerged aquatic vegetation to ameliorate seawater acidity was studied. The authors found that S. latissimi. Had the greatest capacity to ameliorate seawater acidity when pCO2 was experimentally elevated and that S. latissimi was able to take up more CO2/HCO3- to ameliorate seawater acidity as pCO2 increased. I think this is a well thought out study and that the methods are sound. The only thing I take issue with is that the future scenario is a CO2 x warming experiment and I would constantly forget that when reading about experiment 2. I think the text should be edited to clarify this so that the impact that warming 2C has on plant metabolism and gas solubility is better explained.

Abstract: Clear and concise.

Introduction:

This is one of the clearest explanations of ocean acidification I have ever read. The only thing I would change about the introduction is to highlight the interactive impact of changing temperature and CO2. You briefly mention this in line 95, but in the goals (particularly goal 2) you do not directly highlight that this is something tested and the warming effect is not really discussed in the results/discussion. I think it should be made clearer that in experiment 2 you are actually studying a CO2 by warming effect. So not only should plant metabolism increase slightly, but you’re also changing the saturation of dissolved gases in the water slightly.

Methods:

Line 256: Missing % symbol after 0.01.

Results/Discussion:

The results and discussion are well thought out. The only comment the authors should address is below:

Please report the equation, r2, and p-value for each line in figure 1. It seems like there is a much better fit for some species than others for DO especially.

These other comments are just my thoughts. The authors do not need to address them in the text:

It is interesting that there was less of a change in DO under high flow conditions because wouldn’t you expect there to be more photosynthesis occurring with higher flow because of less CO2 limitation or is there a greater change in DO under low flow conditions because it is getting moved out of the system more slowly?

So at current CO2 levels, pH is not being impacted by sugar kelp as much as in low flow environments? How do you think this translates into the natural environment? Are oysters in Casco Bay typically restored in high or low flow areas?

6. PLOS authors have the option to publish the peer review history of their article (what does this mean?). If published, this will include your full peer review and any attached files.

Reviewer #1: No

Reviewer #2: No

---

## [Author Response · Author response to Decision Letter 0]

20 Apr 2023

All this information can be found in the file "Response to Reviewers.pdf" and it is also pasted below:

Editor:

Remark 1: Both the reviewers and I think the manuscript presents novel and important data that will further our understanding of macrophyte impacts on carbonate chemistry, and we are willing to consider a revised version for publication in the journal. Both reviewers brought up some important considerations especially about the impacts of temperature vs CO2 concentration as well as statistical treatments. In addition to these I thought there were two important points that should be discussed. I think that the experiments were done without sediment. It might be important to discuss the potential implications of including below ground tissue in the experiments but having that tissue in the water column. 

Response: The responses to the Reviewers’ remarks about the impacts of temperature vs CO2 concentration and statistical treatments can be found in the responses to each Reviewer below (Remark 2 and 3 from Reviewer 1, and Remark 1 from Reviewer 2). 

All the experiments were done without sediments in order to keep consistency with the set up for all species studied and allow species comparison. The seagrass Zostera marina grows mostly in soft sediments with the belowground tissues usually buried, and the macroalgae species tested grow mostly over rocky bottoms, where the holdfast is exposed to the water column. Therefore, to evaluate the effects of the macrophytes in the water chemistry and compare the responses of the different species, we did not include sediments in the experiment to avoid potential confounding effects from sediment metabolic processes. Nevertheless, as the Editor suggested, there are potential implications of having the seagrass belowground tissue exposed in the water column, such as increased respiration effects, that we have now included in the Discussion section.

The text reads now as:

“although is important to note that seagrass belowground tissue is usually buried in the sediments and having it exposed in the water column as in this study could increase respiration effects in seawater chemistry that should be further studied.”

Remark 2: Further, I think (but am not certain for experiment 2) all incubations were conducted in the light (when macrophytes might ameliorate acidication). I think it would be important to discuss that we aren’t sure about changes in night time respiration under each of these conditions.

Response: Experiment 1 was conducted in the light (at saturating levels), as the main goal was to compare the capacity to ameliorate acidification among the different species during maximum photosynthesis. However, in Experiment 2, one of the aims was to test this amelioration capacity at different light levels, so, there is one step at the beginning of the photosynthesis-irradiance curves generated that was done in the dark with no light. Each step only lasted 1.5h, therefore, this cannot be interpreted as night time respiration, but, it can be interpreted as no light respiration, and as so, we have included a statement on the Discussion section.

The text reads now as:

 “However, these changes might be different during night time, as we have found hints that no-light (i.e. dark) respiration in our experiments could increase acidification in future conditions accordingly to what has been proposed by Pacella et al. 2018.”

Reviewer #1

Remark 1: In Experiment 1, I am not sure if an averaged value for total alkalinity for each species and pCO2 level treatment combination is appropriate to use in estimating other carbonate chemistry parameters used as response variables (line 223). While total alkalinity did not vary much between replicates, using an average value for a subset of replicates within the same treatment combination ‘hides’ within-group variability and may be a form of pseudoreplication. If this cannot be properly justified, it may be more appropriate to run the statistical analyses with the three replicates per treatment combination where total alkalinity was actually measured.

Response: We thank the Reviewer for this thoughtful comment. As stated in the methods for Experiment 1, three representative chambers (out of 4 to 6) from each CO2 level per species were analyzed for total alkalinity before and also after the incubations. As minimal variation was found among chambers, values were averaged and applied to all the chambers for that same species, CO2 level and time of incubation.

Changes in alkalinity as small as we found (from 0 to 8 μmol kg -1) do not translate into relevant changes in the carbonate chemistry parameters calculated in this study (DIC and Ω, as pH was measured in situ). This is, variation observed in total alkalinity values among the three representative chambers translates into small variation in DIC and Ω values, at the point that no variation is found in statistical analysis. Under the conditions in Experiment 1, a change of at least 15 μmol kg -1 (twice what we actually observed) will be needed to reflect changes in DIC and Ω from current reported values that could potentially affect further statistical analysis. Additionally, the variables analyzed in the statistical analysis of this study are pH, DIC and Ω (derived from pH and the averaged total alkalinity). All these variables are independently measured or calculated for each chamber, thus, none of these data hides within-group variability. 

Therefore, as no changes are found and data is independent, if the editor agrees, is our preference to keep the averaged values. We have added details into the small variation found among chambers and how this does not translate into relevant DIC and Ω changes.

The text reads now as:

 “Discrete water samples were acquired from three representative chambers from each CO2 level per species for analysis of total alkalinity (AT) as minimal variation was found among chambers (<10 μmol kg -1) thus not reflecting changes among them.” 

Remark 2: In Experiment 2, the future conditions included increased temperature by 2°C but there is little attention given to the effects of warming throughout the manuscript. It would be helpful to provide context on whether this is a modest increase in temperature in terms of S. latissima physiology and whether 13°C falls within its tolerance range.

Response: We agree with the Reviewer that little attention is given to the increase in temperature by 2°C along the manuscript (from 11°C to 13°C), and this is because the optimum growth of S. latissima sporophytes is reported between 10°C and 15°C (Boltom and Lüning 1982). Therefore, at 13°C the effects of such an increased compared to 11°C should be mild, although we did not specifically test this.

To avoid confusion and provide more context, we have included the physiological thresholds information within the text when describing the study species in the Methods section and also in the Discussion section.

The text reads now as:

“…and represents one of the most farmed seaweed species in the Atlantic ocean [35,36] with optimum growth of sporophytes reported between 10°C and 15°C [37].”

“Our results also suggest that higher availability of CO2, combined with slightly higher temperature (in our case within the range of optimum growth of the species studied), enhances the effect of light on sugar kelp,…”

References:

Bolton, J. J., and Lüning, K. (1982). Optimal growth and maximal survival temperatures of Atlantic Laminaria species (Phaeophyta) in culture. Mar. Biol. 66, 89–94. doi: 10.1007/BF00397259

Remark 3: There is also no mention of whether the elevated temperature in the future conditions in Experiment 2 could have confounded the effects of residence time and pCO2 levels. The treatments were not independent (i.e., there were no treatments with ambient pCO2 + future temperature or treatments with future pCO2 + ambient temperature), so while the ambient and future conditions may reflect the most realistic combinations of pCO2 levels and temperatures, it’s not possible to rule out a temperature effect here.

Response: The research questions we asked in Experiment 2 are aiming at understanding responses of S. latissima in current vs. future scenarios, rather than aiming to assess the role of temperature and/or pCO2 independently. This collapsed design approach is commonly used to answer questions related to IPCC predictions as both warming and temperature will increase in the future (see Boyd et al. 2018). Nevertheless, we agree with the Reviewer that we cannot rule out a temperature effect in our results and as so, we have included text explaining the potential effects of temperature in the Discussion section. See also Remark 1 by Reviewer 2.

The text reads now as:

 “Yet, the slight increase in temperature in the future conditions’ treatment can decrease solubility of CO2 as well as increase the macrophyte metabolic activity which can limit its availability.”

References:

Boyd, P. W., Dillingham, P. W., McGraw, C. M., Armstrong, E. A., Corn- wall, C. E., Feng, Y. y., ... Nunn, B. L. (2015). Physiological responses of a Southern Ocean diatom to complex future ocean conditions. Nature Climate Change, 6, 207–213. https://doi.org/10.1038/NCLI MATE2811

Remark 4: The predictive model is a bit unclear, so it may be helpful to provide a written formulation of the model. In some parts of the manuscript, the model seems to include biomass as a predictor variable (line 39) but in others, the response variable appears to be standardized by biomass (lines 294-295). If the response variable is change in DIC per gram of biomass, then biomass is not really a predictor in the model but rather the unit of measurement for the response variable. Scaling the response variable by biomass relies on the assumption that metabolism (e.g., DIC uptake) scales linearly with biomass.

Response: As suggested by the Reviewer, we have provided written formulation of the model to avoid potential confusion. As stated by the Reviewer, biomass is not a predictor on the model, but the unit of measurement for the response variable (e.g. DIC per gram of biomass). Scaling the response variable by biomass relies on the assumption that metabolism scales linearly with biomass, which has been seen on results from Ricart et al. and Wahl et al. 2018 with other marine macrophyte species. However, to our knowledge the slope of this linear relation for S. latissima has not yet been assessed and hence, the model on its current form cannot include it.

We have added a statement in the Discussion section reflecting this assumption and the need for further studies.

The text reads now as:

“Linear relation among biomass and metabolically linked seawater chemistry responses has been demonstrated in [14,75] for other marine macrophyte species, but not yet for sugar kelp.”

References:

Ricart AM, Gaylord B, Hill TM, Sigwart JD, Shukla P, Ward M, et al. Seagrass-driven changes in carbonate chemistry enhance oyster shell growth. Oecologia. 2021;196: 565–576. doi:10.1007/s00442-021-04949-0

Wahl M, Schneider Covachã S, Saderne V, Hiebenthal C, Müller JD, Pansch C, et al. Macroalgae may mitigate ocean acidification effects on mussel calcification by increasing pH and its fluctuations. Limnol Oceanogr. 2018;63: 3–21. doi:10.1002/lno.10608

Remark 5: There could be more discussion on the ecosystem-level implications of the laboratory-based findings. In natural settings, seawater acidity amelioration by macrophytes likely varies over space and time. The authors touched on how basin geomorphology, diurnal cycles, and water clarity will affect the ability of S.latissima to ameliorate seawater acidity. If S.latissima produces the greatest amelioration effects under saturating light conditions compared to the other species, are these effects countered by stronger respiration effects, potentially increasing seawater acidity, under dark conditions? If so, I think it is important to caveat that S.latissima may not be the ideal candidate for acidification mitigation. Additionally, other processes occurring in the water column or benthos (e.g., alkalinity production/consumption related to respiration, air-water CO2 exchange, etc.) can affect acidity. How might these uncertainties affect the reliability of the predictive model?

Response: We agree with the Reviewer’s remark. In fact, most of the processes that can affect seawater chemistry mentioned above were already included in the Introduction of the manuscript and should be considered when extrapolating these results into natural ecosystems. The Reviewer also asks a very interesting question: “If S. latissima produces the greatest amelioration effects under saturating light conditions compared to the other species, are these effects countered by stronger respiration effects, potentially increasing seawater acidity, under dark conditions?” this is something we did not tested in this study, but our data shows hints that no-light (i.e. dark) respiration could increase acidification based on the differences seen from the control treatments. However, this should be further studied as the fact that S. latissima shows amelioration of acidification when exposed to light saturating conditions does not necessarily have to translate into the opposite under dark conditions. For instance, in Ricart et al. 2021 a biomass gradient of the seagrass Zostera marina under light conditions showed an increase in pH levels with an increase in biomass, but this was not the case under dark conditions were all biomass levels in the gradient, including a control with no biomass, showed the same pH levels. Similar results were shown by Wahl et al. 2018 with macroalgae of the Fucus genera.

As recommended by the Reviewer we have included more Discussion on the ecosystem-level implications of the laboratory-based findings, and have included a statement on how these uncertainties can affect the reliability of the predictive model, as well as the need for further studies under dark conditions. 

The text reads now as:

 “However, these changes might be different during night time, as we have found hints that no-light (i.e. dark) respiration in our experiments could increase acidification in future conditions accordingly to what has been proposed by [74].”

“Nevertheless, these findings are limited to a laboratory setting in the absence of air-water exchange and more studies in the field are crucial to understand the biological and physical drivers of coastal CO2 stoichiometry.”

References:

Ricart AM, Gaylord B, Hill TM, Sigwart JD, Shukla P, Ward M, et al. Seagrass-driven changes in carbonate chemistry enhance oyster shell growth. Oecologia. 2021;196: 565–576. doi:10.1007/s00442-021-04949-0

Wahl M, Schneider Covachã S, Saderne V, Hiebenthal C, Müller JD, Pansch C, et al. Macroalgae may mitigate ocean acidification effects on mussel calcification by increasing pH and its fluctuations. Limnol Oceanogr. 2018;63: 3–21. doi:10.1002/lno.10608

Remark 6: line 106: it may be helpful to define residence time.

Response: As suggested by the Reviewer we have defined residence time.

The text reads now as:

 “…by affecting water residence time (i.e., how fast water moves through a system in equilibrium).”

Remark 7: lines 130-132: alternation of generations is not particularly useful to include here since it is common in most plants and algae to some degree.

Response: We have removed this information in the text.

Remark 8: lines 149-150: how did you ensure equilibrium was reached before starting the experiment?

Response: Yes, we did. The CO2 was manipulated by continuously bubbling pre-mixed compressed air into each treatment chamber from the bottom to equilibrate the water to the desired treatment (for 48h, during the acclimation period). We ensured equilibrium by monitoring the pH/O2 in the chambers and observing that they reached a steady state with a hand-held sensor (HQ40D, Hach Lange). 

We have added this information into the Methods section. The text reads now as:

“CO2 was manipulated by bubbling pre-mixed compressed air from tanks at the desired concentrations into each treatment chamber from the bottom to equilibrate the water to the desired treatment and checking equilibrium by measuring constant pH and DO with a hand-held sensor (HQ40D, Hach Lange).”

Remark 9: lines 153-154: what was the flow rate?

Response: We had tanks of pre-mixed air with the concentration that were used for each of the pCO2 treatments. Therefore, the exact flow rate in the chambers was not measured, but was set up with the same pressure exit from the tanks for all of them (1bar). During the initial setup process, we adjusted the tank manifold settings to have roughly equal flow across all chambers and treatments (manifold reduced pressure to 1bar from the tank to its entrance in the chambers). We then used the pH/O2 probe to monitor the conditions within the chambers and ensure that conditions were being maintained. See also Remark 8 by the same Reviewer.

We have added this information into the Methods section. The text reads now as:

“CO2 was manipulated by bubbling pre-mixed compressed air from tanks at the desired concentrations into each treatment chamber from the bottom to equilibrate the water to the desired treatment and checking equilibrium by measuring constant pH and DO with a hand-held sensor (HQ40D, Hach Lange).”

Remark 10: lines 164-165: how did temperature variation throughout the incubation affect carbonate chemistry calculations?

Response: We would like to clarify that the temperature did not change along each incubation. Therefore, no effect is expected in the carbonate chemistry calculations and hence comparison from the samples analyzed before and after the incubations. 

Remark 11: lines 166: maybe state upfront the measured carbonate chemistry parameters.

Response: Done as requested by the Reviewer. The text reads now as:

“Initial and final measurements of seawater carbonate chemistry parameters (pH and total alkalinity, see details below) …”

Remark 12: line 179: maybe explain that the pH controller system could maintain a pCO2 level within a range of 200 μatm.

Response: Done as requested by the Reviewer. The text reads now as:

“where CO2 was bubbled into seawater using solenoid valves and a Neptune Apex Controller pH system (feedback sensors were in the mixing tanks; resolution <200 μatm pCO2)”

Remark 13: lines 199-200: how did you ensure equilibrium was reached before starting the experiment?

Response: The CO2 was injected through dispersion into fine bubbles in an in-line equilibrator and delivery and equilibrium was controlled by a solenoid valve and pH monitoring control (Neptune Apex Controller pH system). The CO2 injection by solenoid control and pH monitoring feedback to trigger the solenoid programming was tested for 1 month prior to the experiment to determine effective solenoid limits for on/off starts to ensure desired pH value maintenance on daily and weekly time scales.

Remark 14: lines 284-286: it’s not clear how the same light curves approach was applied to the other response variables. In equations (1) and (2), were P and Pmax terms simply replaced by net ΔDIC/ΔpH/ΔΩ between tank inflow and outflow and the maximum ΔDIC/ΔpH/ΔΩ at saturating irradiance, respectively? I suggest writing out the equations in a more generalized form for clarity. For example, P and Pmax could be replaced with other terms, such as R and Rmax, and state that R is ΔDO/ΔDIC/ΔpH/ΔΩ.

Response: Done as requested by the Reviewer. The text reads now as:

“Thus, ΔDO in P and Pmax terms can be replaced by ΔDIC/ΔpH/ΔΩ.”

Remark 15: line 292: was leaf surface area accounted for as well?

Response: Prior to the experiment, we collected data of surface area and biomass of the S. latissima thallus. A linear relation exists between them (see below) which shows that the same species, same biomass and same surface area was used across treatments in Experiment 2.

Remark 16: line 305: it’s best practice to state R version used for analyses.

Response: Done as requested by the Reviewer. The text reads now as:

“…statistical analysis presented were done in the R software (R 4.2.1)”

Remark 17: lines 328-330: should the values mentioned here match the slopes presented in Table S3? If not, where do these numbers come from?

Response: The linear model is constructed as a lm() in R, and the output of the model, which appears using function summary(), reports the model exactly as we show in the Table S3. This includes the intercept of each species and the slopes of each species. To avoid any confusion, we have added a clarification on the information shown in the caption of Table S3 and how to calculate the slopes for each species. See also Remark 18 and 25 by the same Reviewer.

The text reads now as:

“Summary output from linear models as reported in the R software. Estimates and SE are provided, where SE is within (). From these estimates, the intercepts and slopes of each linear interaction can be calculated. Intercepts of each species should be calculated by adding them to the intercept of the control (row Intercept). Slopes of each species should be calculated by adding them to the slope of the control (row pCO2).”

Remark 18: lines 334-339: should the values mentioned here match the slopes presented in Table S3? If not, where do these numbers come from?

Response: Table S3 values do not show the slopes, but show all the information needed to calculate them. See also Remark 17 and 25 by the same Reviewer.

Remark 19: lines 357-359: was the higher impact of kelp in future compared ambient conditions assessed qualitatively?

Response: As stated by the Reviewer, this comparison is done qualitatively based on the control vs. sugar kelp patterns that appear in Figure 2.

Remark 20: lines 446-447: since only two flow rates (0.02 and 0.06 cm s-1) were used to parameterize the model, are there limitations when interpreting extrapolated flows an order of magnitude higher?

Response: We have not experimentally tested other flow rates. Therefore, despite results with flow rates an order of magnitude higher could be extrapolated from the model, the interpretation should be done with caution. More studies at higher flow rates are needed to answer this question, and most probably, at much higher flow rates the effects of marine macrophytes in water chemistry will be minimal.

Remark 21: line 469: in the example starting on line 469, the predictive model is applied to an example. The spatial extent of the effect is noted (1000 m3), but what about the temporal extent, i.e., how long would the effect be sustained?

Response: As stated by the Reviewer we use an example to illustrate the use of the model. However, the volume chosen does not reflect the spatial extent of the amelioration effect. Instead, it reflects a known volume of water to which the model is applied. The temporal extent, is a great question that is far beyond the scope of the present effort. In fact, further research is on the track of studying how the marine macrophytes OA amelioration extends to larger spatial and temporal scales (Ricart et al. unpublished).

Remark 22: lines 496-503: mention that findings are limited to macrophyte seawater acidity amelioration during photosynthesis in a laboratory setting, in the absence of air-water exchange, etc.

Response: The mention has been added as suggested by the Reviewer.

The text reads now as:

 “Nevertheless, these findings are limited to a laboratory setting in the absence of air-water exchange and more studies in the field are crucial to understand the biological and physical drivers of coastal CO2 stoichiometry.”

Remark 23: Table 1: label experiments numerically, e.g., ‘Experiment 1 – Comparison among species’ and include statistical analyses used, either in the table or table caption. If ANOVA was used to compare overall effects of species, pCO2, and their interaction for Experiment 1, this should be listed in the methods, too.

Response: Table 1 is now labelled as suggested by the Reviewer. Just for clarification, the linear model applied was not an ANOVA (as pCO2 is a continuous variable).

Remark 24: Table S1: since the number of replicates per treatment combination (n) varied, consider including n for each treatment combination in this table.

Response: We have included the number of replicates on each treatment in Table S1.

Remark 25: Table S3: how was the linear model constructed so that coefficients were available for each species? I understand that an intercept-free model in lm() can accomplish this, but an intercept is reported in this table. More description in the methods would make the data analyses more reproducible.

Response: The linear model is constructed as a lm() in R, and the output of the model, which appears using function summary(), reports the model exactly as we show in the Table S3 with an intercept for each species and a the control treatment. To avoid any confusion, we have added a clarification on the information shown in the caption of Table S3 and how to calculate the slopes for each species. See also Remark 17 and 18 by the same Reviewer.

The text reads now as:

“Summary output from linear models as reported in the R software. Estimates and SE are provided, where SE is within (). From these estimates, the intercepts and slopes of each linear interaction can be calculated. Intercepts of each species should be calculated by adding them to the intercept of the control (row Intercept). Slopes of each species should be calculated by adding them to the slope of the control (row pCO2).”

Reviewer #2: 

Remark 1: In this study the capacity of submerged aquatic vegetation to ameliorate seawater acidity was studied. The authors found that S. latissimi. Had the greatest capacity to ameliorate seawater acidity when pCO2 was experimentally elevated and that S. latissimi was able to take up more CO2/HCO3- to ameliorate seawater acidity as pCO2 increased. I think this is a well thought out study and that the methods are sound. The only thing I take issue with is that the future scenario is a CO2 x warming experiment and I would constantly forget that when reading about experiment 2. I think the text should be edited to clarify this so that the impact that warming 2C has on plant metabolism and gas solubility is better explained.

Response: We thank the Reviewer for this constructive comment. We have edited the text in the Discussion to clarify that the increase in 2°C can affect plant metabolism (see also Remark 3 by Reviewer 1) and also gas solubility. 

The text reads now as:

“Yet, the slight increase in temperature in the future conditions’ treatment can decrease solubility of CO2 as well as increase the macrophyte metabolic activity which can limit its availability.”

Remark 2: Abstract: Clear and concise.

Response: Thank you.

Remark 3: Introduction: This is one of the clearest explanations of ocean acidification I have ever read. The only thing I would change about the introduction is to highlight the interactive impact of changing temperature and CO2. You briefly mention this in line 95, but in the goals (particularly goal 2) you do not directly highlight that this is something tested and the warming effect is not really discussed in the results/discussion. I think it should be made clearer that in experiment 2 you are actually studying a CO2 by warming effect. So not only should plant metabolism increase slightly, but you’re also changing the saturation of dissolved gases in the water slightly.

Response: We have added made clearer in the Introduction and Goals that the Experiment 2 is studying a CO2 by warming effect. See also Remark 2 by the same Reviewer.

The text reads now as:

“…to (2) assess the effects of water residence time on its amelioration capacity in ambient and simulated future scenarios of climate change (increased CO2 and temperature)…”

Remark 4: Methods: Line 256: Missing % symbol after 0.01.

Response: The symbol as been added as suggested by the Reviewer. 

The text reads now as:

“…and for Ω 0.01 %.”

Remark 5: Results/Discussion: The results and discussion are well thought out. The only comment the authors should address is below:

Please report the equation, r2, and p-value for each line in figure 1. It seems like there is a much better fit for some species than others for DO especially.

Response: We would like to clarify that we did not run a simple regression analysis for each species in Experiment 1, and as a result, the equation, r2, and p-value for each line in Figure 1 was not reported. This is because our goal was to check for differences among species on the effects of pCO2, instead of checking the relation between release of DO, uptake of DIC or change in pH and Ω with pCO2 for each of the species. 

Figure 1 includes lines and shaded areas which, as is stated in the figure caption, represent predicted average values (lines) and the 95% Confidence Interval (shade areas) calculated from the estimates provided by the statistical model applied. The results and estimates of the model are provided in Table 1 and Table S3 respectively. We could include the equations on each line, but to avoid confusion with a simple regression analysis, is our preference to keep Figure 1 in its current form.

Remark 6: These other comments are just my thoughts. The authors do not need to address them in the text:

It is interesting that there was less of a change in DO under high flow conditions because wouldn’t you expect there to be more photosynthesis occurring with higher flow because of less CO2 limitation or is there a greater change in DO under low flow conditions because it is getting moved out of the system more slowly?

So at current CO2 levels, pH is not being impacted by sugar kelp as much as in low flow environments? How do you think this translates into the natural environment? Are oysters in Casco Bay typically restored in high or low flow areas?

Response: We agree with the Reviewer that most likely there is a greater change in DO under low flow conditions because it is getting moved out of the system more slowly, and similarly happens with DIC and H+, and hence the results obtained for pH and Ω. This could be translated as more hydrodynamics more dispersal of the chemical signal and less effect of the sugar kelp metabolism on the amelioration of ocean acidification, which has been pointed out in other works, such as for instance Noisette et al. 2022. 

Casco Bay has oyster farms for the aquaculture industry, as well as oyster reef restoration projects. Despite we do not know if these farms or projects are considering flow in their designs, to install these farms or projects in low flow areas would likely increase rates of success. However, based on our results, this might not be relevant in future conditions.

References:

Noisette F, Pansch C, Wall M, Wahl M, Hurd CL. Role of hydrodynamics in shaping chemical habitats and modulating the responses of coastal benthic systems to ocean global change. Glob Chang Biol. 2022;28: 3812–3829. doi:10.1111/gcb.16165

---

## [Decision Letter · Decision Letter 1]

29 Jun 2023

Optimizing marine macrophyte capacity to locally ameliorate ocean acidification under variable light and flow regimes: Insights from an experimental approach

PONE-D-22-32214R1

Dear Dr. Ricart,

We’re pleased to inform you that your manuscript has been judged scientifically suitable for publication and will be formally accepted for publication once it meets all outstanding technical requirements.

Kind regards,

Laura Reynolds

Academic Editor

PLOS ONE

Additional Editor Comments (optional):

Thank you for your careful consideration of reviewer suggestions. The reviewer and I both agree that this is a well supported, intersting paper. Congratulations.

Reviewers' comments:

Reviewer's Responses to Questions

**Comments to the Author**

1. If the authors have adequately addressed your comments raised in a previous round of review and you feel that this manuscript is now acceptable for publication, you may indicate that here to bypass the “Comments to the Author” section, enter your conflict of interest statement in the “Confidential to Editor” section, and submit your "Accept" recommendation.

Reviewer #1: All comments have been addressed

2. Is the manuscript technically sound, and do the data support the conclusions?

Reviewer #1: Yes

3. Has the statistical analysis been performed appropriately and rigorously? 

Reviewer #1: Yes

4. Have the authors made all data underlying the findings in their manuscript fully available?

Reviewer #1: Yes

5. Is the manuscript presented in an intelligible fashion and written in standard English?

Reviewer #1: Yes

6. Review Comments to the Author

Reviewer #1: I thank the authors for responding to and clarifying all of my questions/concerns, especially the methodological/statistical questions that went a bit into the weeds. Congratulations on an interesting and very timely study!

7. PLOS authors have the option to publish the peer review history of their article (what does this mean?). If published, this will include your full peer review and any attached files.

Reviewer #1: No

---

## [Editor Report · Acceptance letter]

15 Sep 2023

PONE-D-22-32214R1 

Optimizing marine macrophyte capacity to locally ameliorate ocean acidification under variable light and flow regimes: Insights from an experimental approach 

Dear Dr. Ricart:

I'm pleased to inform you that your manuscript has been deemed suitable for publication in PLOS ONE. Congratulations! Your manuscript is now with our production department. 

Kind regards, 

on behalf of

Dr. Laura Reynolds 

Academic Editor

PLOS ONE